# Small Cell Transformation of EGFR-Mutant NSCLC Treated with Tyrosine Kinase Inhibition

**DOI:** 10.3390/curroncol32100554

**Published:** 2025-10-03

**Authors:** Adam Rock, Isa Mambetsariev, Siddhika Pareek, Jeremy Fricke, Xiaochen Li, Javier Arias-Romero, Waasil Kareem, Leonidas Arvanitis, Debora S. Bruno, Stacy Gray, Ravi Salgia

**Affiliations:** 1Department of Medical Oncology & Therapeutics Research, City of Hope Comprehensive Cancer Center, Duarte, CA 91010, USAspareek@coh.org (S.P.);; 2Division of Biostatistics, City of Hope Comprehensive Cancer Center, Duarte, CA 91010, USA; 3Department of Medicine, City of Hope Comprehensive Cancer Center, Duarte, CA 91010, USA; wkareem@coh.org; 4Department of Pathology, City of Hope Comprehensive Cancer Center, Duarte, CA 91010, USA; larvanitis@coh.org; 5Department of Medical Oncology & Therapeutics Research, City of Hope Cancer Center Atlanta, Newnan, GA 30265, USA; dbruno@coh.org

**Keywords:** EGFR, non-small cell lung cancer, transformation, small cell lung cancer, germline, somatic

## Abstract

**Simple Summary:**

Histologic transformation represents the conversion of a baseline cancerous cell into a new cell, leading to resistance to specific treatments. Small cell transformation is an example of resistance to tyrosine kinase inhibition in epidermal growth factor mutant lung cancer. This retrospective review focuses on genomic alterations at various time points observed in patients with epidermal growth factor mutant lung cancer who underwent histologic transformation. We observed various recurrent genomic alterations that may contribute to the development of small cell lung cancer in the setting of targeted therapy for EGFR mutations. Relevant data pertaining to outcomes and therapeutic interventions are described in detail to aid clinicians in addressing this rare and clinically challenging phenomenon.

**Abstract:**

Introduction: Epidermal growth factor receptor (EGFR) alterations exist in 15–50% of non-small cell lung cancer (NSCLC) diagnoses. Although effective therapeutics have been developed in the form of tyrosine kinase inhibitors (TKI), various mechanisms of resistance lead to treatment failure after exposure to EGFR TKI-based therapy. Of these, histologic transformation (HT) into small cell lung cancer (SCLC) represents approximately 14% of cases. Methods: Within a single institution, we retrospectively reviewed longitudinal data from both tissue and liquid biopsies of patients with histologic transformation after a diagnosis of EGFR-mutant NSCLC. We sought to further characterize the baseline and emergent genomic alterations after HT to SCLC in the context of TKI exposure, along with germline alterations that may contribute to lineage plasticity and outcomes. Results: Fifteen patients were included in our analysis. Of these, EGFR exon 19 deletions were the most frequent (*n* = 11, 73.3%), followed by L858R (*n* = 3, 20%) and L861Q (*n* = 1, 6.7%). The median time for transformation was 17 months (95%CI, 8.9–41.9 months). The median OS of our cohort was 51.6 months (95%CI, 26.3—NE) with a median OS post-transformation of 13.4 months. Recurrent genomic alterations included TP53, Rb1, PIK3CA, and BRAF. Germline testing revealed a pathogenic alteration in FBN1, with a recurrent variant of unknown significance (VUS) in PALLD. Conclusion: Post-transformation somatic mutation testing and germline testing at presentation revealed unique mutational profiles not previously reported in the setting of HT to SCLC. Further investigations are required to determine the optimal treatment and sequencing following HT.

## 1. Introduction

Non-small cell lung cancer (NSCLC) accounts for most annual lung cancer diagnoses and remains the most frequent cause of cancer-related deaths [1]. The incorporation of routine comprehensive genomic profiling (CGP) has elucidated molecular driver alterations that have led to significant advances in therapeutic approaches. Of these, alterations within the epidermal growth factor receptor (*EGFR*) represent the most frequent event estimated to occur in 15–50% of NSCLC patients [2]. Therapeutic strategies using tyrosine kinase inhibitors (TKIs) have been developed to combat aberrant signaling within the *EGFR* pathway. Despite the advent of molecular targeted therapies, acquired resistance inevitably develops. After secondary resistance alterations, histological transformation (HT) is the second most common etiology of acquired resistance, affecting up to 14% of patients [3]. HT represents a poorly understood mechanism of resistance in which a patient’s original malignant histology undergoes transformation into a distinct, new histologic type. Various examples of HT have been described in NSCLC including squamous cell carcinoma, large cell neuroendocrine carcinoma, and small cell lung cancer (SCLC). Whether this form of resistance is secondary to acquired HT, or rather, selective pressure on a mosaic baseline histology remains unclear.

Notably, HT does not appear to be unique to *EGFR*-mutant NSCLC. HT has been reported in NSCLC patients with and without targetable mutations. Reports of lineage plasticity have been documented in *KRAS G12C*-, *ROS1*-, and *ALK*-altered NSCLC after targeted therapy [4,5,6,7,8,9]. Interestingly, HT also appears to be a resistance mechanism in wild-type NSCLC after exposure to immune checkpoint blockade [10,11,12]. Furthermore, HT does not appear exclusive to NSCLC with cases in prostate and bladder cancer after exposure to cancer-directed therapy [13,14].

Despite the efficacy of TKI therapy, *EGFR*-targeted therapies appear to be suppressive rather than curative, allowing for quiescent cells to ultimately develop resistance [15,16]. At the time of progression to *EGFR*-targeted TKI therapy, histologic transformation in *EGFR*-mutant lung adenocarcinoma (LUAD) is observed in 5–15% [2,3]. Investigation into the continued expression of *EGFR* protein following transformation has demonstrated conflicting results [2,17].

Multiple retrospective analyses have worked to characterize the genomic landscape of transformed *EGFR*-mutant SCLC. Recurrent alterations in *TP53*, *RB1*, and *PIK3CA* have been previously reported [2]. Additionally, patients harboring baseline *EGFR*-mutant LUAD and concurrent *RB1* and *TP53* loss were found to be at high risk (18%) of HT relative to those with wild type *RB1* and *TP53* status (0%) [18,19]. While *RB1* loss is frequently observed, it does not appear to be the only necessary event to induce histologic transformation in *EGFR*-mutant LUAD [18]. Furthermore, transformed SCLC cases were enriched with various molecular pathway alterations, including *MAPK*, *Jak-STAT*, *ERBB*, *FGFR*, mechanistic targeting of rapamycin kinase signaling, and *PI3K-AKT-mTOR* pathways [19].

As genomic alterations of *TP53* and *RB1* loss do not appear to be the only required “hit” to elicit HT, we postulate that there may be additional alterations at the genomic, genetic, or epigenetic levels that promote HT. To further characterize the genomic and genetic alterations contributing to histological transformation, we retrospectively reviewed all molecular and genetic data observed in patients treated at the City of Hope Comprehensive Cancer Center.

## 2. Methods

We reviewed the longitudinal data of both tissue and liquid biopsies of patients experiencing histological transformation after a diagnosis of *EGFR*-mutant NSCLC. We selected a total of 15 patients (*n* = 15) who had an initial diagnosis of *EGFR*-mutant NSCLC and, upon treatment, transformed to histologically confirmed SCLC between 2014 and 2023. Patient data was deidentified and as per City of Hope guidelines with City of Hope Institutional Review Board approval under IRB#21458 and IRB#21199 and in accordance with the Declaration of Helsinki. Genomic data were collected through a retrospective chart review of clinical information, performed by the primary treating physician in the context of routine clinical care. Liquid and tissue biopsy molecular testing was performed as part of standard of care by the primary oncologist using internal and commercially available next-generation sequencing platforms including the City of Hope HopeSeq, Foundation Medicine, Guardant Health, Clarient Diagnostic, QUEST Diagnostic, and Mayo Clinic Laboratories. Germline testing was performed under IRB#21199, evaluating longitudinal outcomes of germline testing of patients undergoing paired tumor-normal WES, cancer susceptibility testing, and testing for actionable genetic conditions. Treatment outcomes including progression free survival (PFS) and overall survival (OS) were calculated by Kaplan–Meier method.

## 3. Results

Fifteen patients were included in the study. As depicted in Table 1, the median age was 60 years (IQR 54.5–66.5). Gender distribution was similar with 53% male (*n* = 8) and 47% female (*n* = 7) patients. The majority of patients were Asian 60% (9/15) and had never smoked with or without passive exposure (60%) (9/15). *EGFR* exon 19 deletion was the most common *EGFR* subtype at 73.3% (*n* = 11), followed by *EGFR* L858R (*n* = 3, 20%), and 1 patient had an L861Q mutation in exon 21 (6.7%). The median number of treatments prior to HT was one (range 1–4).

As illustrated in Figure 1A, the median time to transformation was 17 months (95%CI, 8.9–41.9 months). The median OS of our cohort was 51.6 months (95%CI, 26.3—NE) with a median OS post-transformation of 13.4 months (Figure 1B, 95%CI, 7.9—NE). Unsurprisingly, progression-free survival (PFS) of 1st line therapy was 15.2 months (95%CI, 5.9, 23.6), consistent with the known efficacy of EGFR TKIs prior to transformation. Excluding patients undergoing HT while on 1st line therapy, we observed a median time to transformation of 6.0 months (range 0–44.4 months). The median PFS of 1st line therapy was 15.2 months (95%CI, 5.9–23.4) (Figure 1C). With 2nd and 3rd line therapy, we observed median PFS of 5.6 (95%CI, 3.8–25.1) and 4.6 months (95%CI, 2.9–7.0), respectively.

All patients had canonical *EGFR* alterations, including exon 19 deletion (*n* = 11, 73.3%), L858R mutation (*n* = 3, 20%), or L861Q mutation (*n* = 1, 6.7%) (Figure 2). Co-occurring EGFR alterations occurred in the form of EGFR amplification (*n* = 4, 26.7%) and C797s mutation (*n* = 1, 6.7%). There were no observed alterations with EGFR T790M. None of the patients had changes in the canonical driver alterations over the course of transformation and subsequent treatment. The most common co-occurring genomic alteration was *TP53*, which was observed in 14 patients (93%). Additional coalterations were observed in *PIK3CA* (*n* = 8, 53%), *RB1* (*n* = 6, 40%), *CCNE1* (*n* = 3, 20%), and *CDKN2B* (*n* = 3, 20%). The *PIK3CA/PTEN/AKT* pathway was altered in a high proportion of patients (*n* = 11, 73%). BRAF alterations were also common, represented by rearrangement (*n* = 1), fusion (*n* = 1), and amplification (*n* = 1). *BRCA2* alterations were the most common VUS (*n* = 4, 26.67%) and all were detected at presentation.

Figure 3 shows the intersection of genomic alterations over time between the first and second genomic testing reports. The majority of patients had a limited number of co-mutations at the initial genomic testing report. RB1 and TP53 mutations were the most commonly acquired alterations at the second time-point. FGF10, RICTOR, and AKT1 were the pathogenic mutations detected at the second molecular testing time point.

Five patients underwent germline testing as part of the institutional germline testing protocol, and germline alterations were detected in three patients, as depicted in Figure 4. Our analysis revealed only one known pathogenic alteration in *FBN1* (*n* = 1; 33%). Variants of unknown significance (VUS) were noted as follows: *PALLD* (*n* = 2), *ATR* (*n* = 1), *BLM* (*n* = 1), *BRCA2* (*n* = 1), *CDH1* (*n* = 1), *CDK4* (*n* = 1), *RAD50* (*n* = 1), *EXT2* (*n* = 1), *CFTR* (*n* = 1), *ERCC4* (*n* = 1), *FANCI* (*n* = 1), *GEN1* (*n* = 1), and *NTHL1* (*n* = 1). The presence of a germline alteration could not be correlated with outcomes due to the limited sample size; however, all three patients had comparable overall survival.

All HT events were observed during exposure to TKI-based therapy. The median number of lines of therapy was four (range 2–6) with all patients receiving at least one line of EGFR TKI therapy (Figure 5). 26 individual drug therapies and combination therapies were administered in this cohort. The shortest period on TKI therapy was 3 months and the longest was 30 months. Most notably patient 3, who also had a PTEN loss detected after transformation, survived for 58 months after the transformation with 33 months on carboplatin/etoposide. Patients 5 and 12 had the shortest survival times after transformation of only 3 and 2 months, respectively. All patients had at least two or more molecular testing results by either tissue or liquid NGS. Erlotinib and Osimertinib were the most common EGFR TKIs with 10 patients and 11 patients receiving either, respectively. Carboplatin-Etoposide therapy, alone or in combination with EGFR TKI or immunotherapy, was given to all patients after transformation, with a median PFS of 4 months (range 2–32).

## 4. Discussion

The genomic findings are in line with previous analyses that demonstrated a high incidence of TP53, RB1, and PIK3CA alterations [2,18]. However, it does not appear that TP53 and RB1 co-alteration is a strict requirement for transformation, as 46% of the patients (*n* = 7) in our cohort did not exhibit concurrent alterations in either gene. Of note, each commercial panel included assessment of both TP53 and RB1 genes. This finding is recapitulated from previous work by Marcoux et al. [18]. Interestingly, alterations in both *PI3KCA* (*n* = 8, 53%) and the *PIK3CA/PTEN/AKT* pathway (*n* = 11, 73%) were more frequently observed than *RB1* alterations. This enrichment in *PIK3CA* alterations has also been previously observed in a large cohort of EGFR-mutated with SCLC transformation patients when compared to *EGFR* wild-type SCLC patients [20]. Although PIK3CA alterations were enriched in our cohort, RB1 loss alone may not fully encapsulate the extent of RB1 disruption in these cases. Of note, these data are purely descriptive in nature given the limitations of the study design.

*RB1* disruption has been further characterized in an independent cohort of lung adenocarcinoma (LuAD) tumors, which identified RB1 alterations in 14% of cases, with intragenic complex rearrangements (ICRs) that entail a restructuring of a segment of DNA within a single gene comprising the majority of *RB1*-altered tumors (6/7). Notably, RB1 inactivation was significantly more frequent in *EGFR*-mutant tumors (41%) than in EGFR-wild-type tumors (1%) [21]. While our study identified *RB1* loss in 40% of transformed NSCLC cases, we did not assess whether these cases involved ICRs or other structural variations. The prevalence of ICRs in RB1-inactivated *EGFR*-mutant LuADs in an independent cohort suggests that similar mechanisms may exist in other *EGFR*-mutant NSCLC subtypes. However, given the focus of this study on LuADs, further investigation is needed to determine whether this applies to other *EGFR*-mutant NSCLC subtypes and to better understand how *RB1* disruption contributes to lineage switching. In particular, future studies incorporating detailed structural variant analyses of *RB1* in transformed NSCLC cases may help determine whether ICRs contribute to lineage switching and whether they represent a shared mechanism across different histologic transitions, or if NSCLC transformation follows distinct inactivation pathways.

Given the relative enrichment of *PIK3CA* alterations, one would question the utility of the addition of *PIK3CA* inhibitors in these patients. In vitro data of *PIK3CA* inhibitors in combination with *EGFR* TKIs provided some enthusiasm for this approach, with more promise in *PIK3CA* wild-type disease [22]. However, subsequent analyses of PIK3CA inhibitors in combination with the second-generation *EGFR* TKI, erlotinib, have provided modest results [23,24]. The *AKT* inhibitor, capivasertib, is being investigated in combination with osimertinib in TKI naïve, *EGFR* mutant, *PIK3CA/AKT/PTEN altered* patient-derived xenograft models with early success [25]. As *PIK3CA/AKT/PTEN alterations occur frequently in all EGFR-wt NSCLC regardless of HT*, *it is unclear how this may impact a predilection for lineage plasticity*. Interestingly, prior analyses of non-transformed EGFR-mutant NSCLC have demonstrated significantly lower rates of *PIK3CA/AKT/PTEN when compared to this cohort* (14.9% vs. 73%, *respectively*) [25].

Germline analysis was available for 33% of patients (*n* = 5). Germline testing revealed one alteration previously identified as pathogenic (*FBN1*). *FBN1*, found on chromosome 15, encodes for the protein, fibrillin1, which is an essential component of connective tissue formation [26]. *FBN1* germline alterations have been previously associated with Marfan syndrome. Our germline analysis revealed only one recurrent germline alteration, with unclear pathogenicity. A variant of unknown significance was observed in 2 patients (66%) encoding *PALLD*. Interestingly, this gene has been implicated in the stem cell-like characteristics of lung cancer [27]. Our analysis failed to elucidate a recurrent known pathogenic alteration that is correlated with HT. However, this analysis is significantly limited by the sample size and could be further explored in a multi-institutional retrospective fashion.

The time to transformation was 17 months in our study population, which is concordant with previously reported analyses [18]. Compared to previous reports, the OS of our cohort was longer (51.6 months, 95CI% 26.3—NE), which may also be a product of small sample size. Notably, as illustrated in Figure 4, patient 3 showed a durable response to osimertinib alone, following transformation. However, this may be reflective of an atypical indolent disease course, given the long-term survival of this patient post-transformation (>55 months).

One mechanism of resistance that was not evaluated in our analysis was post-translational epigenetic modifications. Unfortunately, we did not have data available to interpret RNA expression as a surrogate for epigenetic modifications. Techniques are being developed to evaluate HT using cell-free DNA assays, which could circumvent or supplement the routine use of tissue biopsy upon progression to targeted therapies [28]. Epigenetic analyses represent an important area for future investigations to delineate additional mechanisms that may contribute to HT.

The efficacy of subsequent treatment requires further investigation with small retrospective data supporting the use of platinum doublet chemotherapy [18]. In our analysis, platinum/etoposide-based chemotherapy combinations were the most frequently used therapies upon progression. Despite the established role of immunotherapies in ES-SCLC and now LS-SCLC, their efficacy in eliciting responses in transformed SCLC has been mixed [18,29]. Although limited by infrequency, the addition of immunotherapy did not appear to provide additional benefits compared to platinum doublet therapy. With the advent of DLL3-targeting agents and recent FDA approval of tarlatamab-dlle, a bispecific T-cell engager targeting DLL3 in ES-SCLC, additional investigation is warranted in EGFR-mutant NSCLC after HT. Available data suggests that DLL3 expression is upregulated upon transformation in EGFR-mutant NSCLC, suggesting that there may be a role for this approach [29]. Unfortunately, given the infrequency and rapidity of HT events, prospective therapeutic trials investigating the utility of ICI seem unlikely to materialize.

Limitations of the present analysis include the small sample size, varying time assessment of molecular testing, and the varying NGS platforms utilized in the context of clinical care. Additionally, there exists potential for overinterpretation of clinical outcomes, including both PFS and OS, given the wide-ranging confidence intervals. Future analyses could utilize a multi-institutional framework to enhance the sample size. Furthermore, while unavailable at the time of this analysis, the institutional non-transformed EGFR mutant NSCLC population could be evaluated to compare to our existing cohort to comment on the rate of HT and frequency of genomic alterations.

## 5. Conclusions

Histological transformation of EGFR-mutated patients into SCLC on EGFR therapy is a complex resistance profile that is becoming more and more common in clinics as patients reach longer survival on newer generation EGFR TKIs. At the same time, the evidence is becoming clear that a simple genomic resistance profile is limited making it difficult to identify patients at risk for transformation and potentially other broader genetic and non-genetic mechanisms need to be further studied. This may include the utilization of germline mutation analysis and further ctDNA analysis. There is also lack of concrete evidence regarding therapy for these patients as noted in our study that patients who re-initiated EGFR TKI alongside chemotherapy may have improved overall survival. Nevertheless, the survival outcomes post-transformation are poor, and new generation therapies such as antibody drug conjugates and novel immunotherapies need to be considered in future clinical trials.

## Figures and Tables

**Figure 1 curroncol-32-00554-f001:**
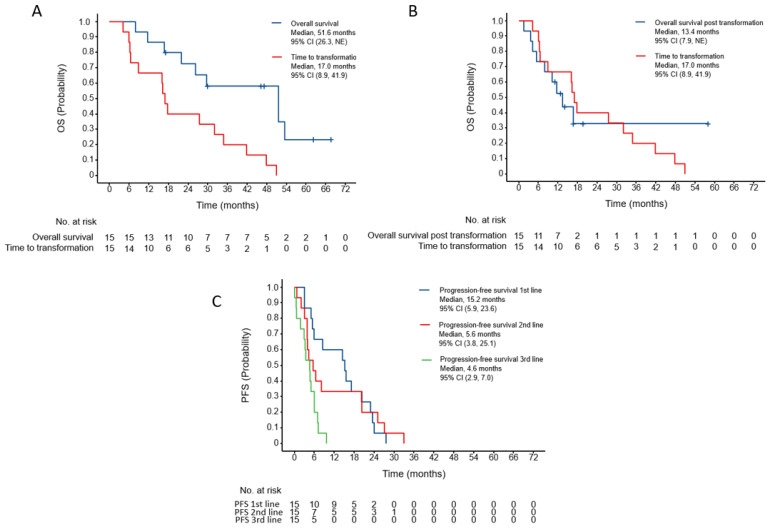
(**A**) Time to transformation versus OS. (**B**) Overall survival post-transformation versus time-to-transformation. (**C**) Progression-free survival by line of therapy.

**Figure 2 curroncol-32-00554-f002:**
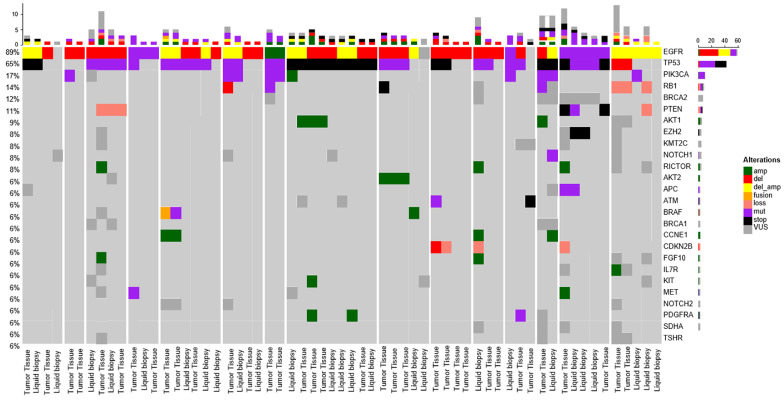
Heat map demonstrating recurrent somatic alterations in patients with transformed EGFR-mutant NSCLC.

**Figure 3 curroncol-32-00554-f003:**
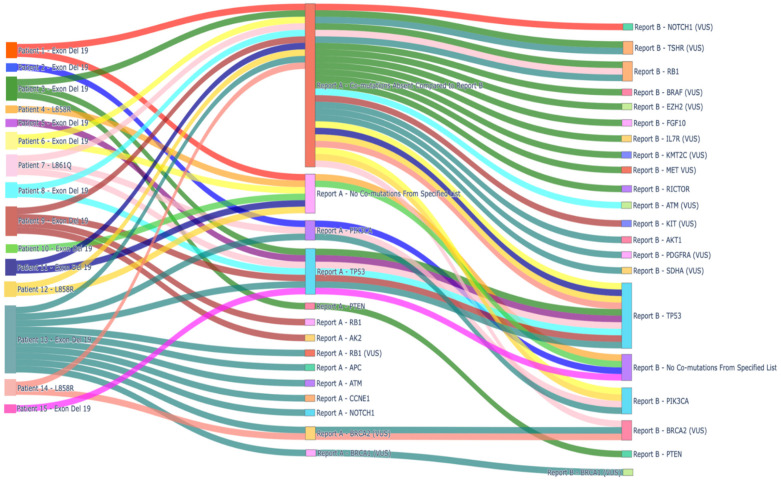
Sankey diagram illustrating the distribution and spectrum of genomic alterations in the cohort. The left nodes represent the driver alteration upon diagnosis. The middle nodes are co-alterations observed at the first genomic assessment chronologically, irrespective of the time of histologic transformation. The right nodes represent additional co-alterations observed after the first assessed time point.

**Figure 4 curroncol-32-00554-f004:**
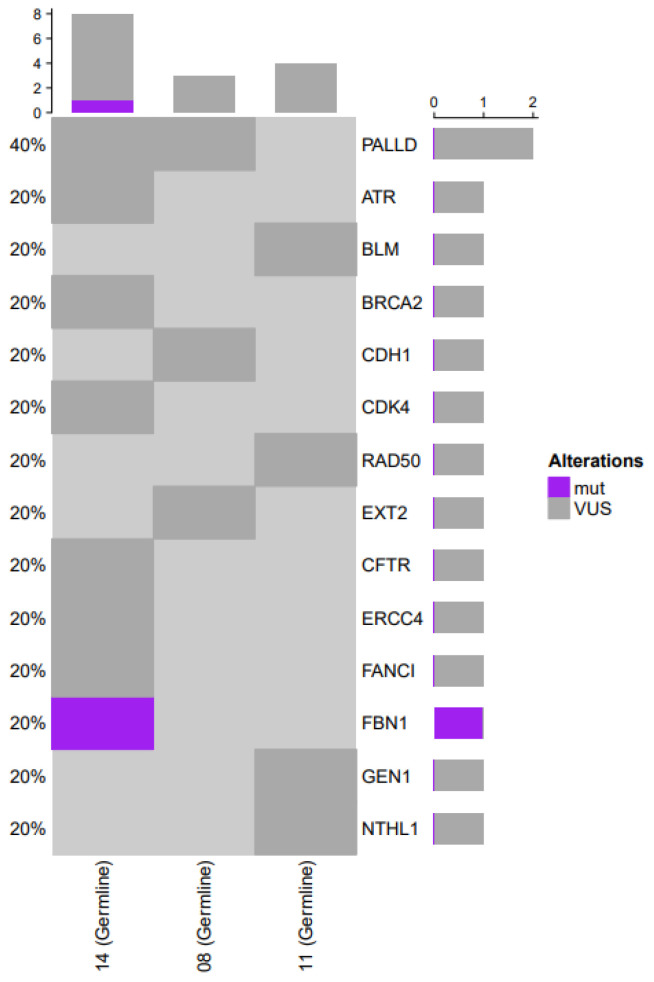
Heat map demonstrating the distribution of germline alterations in EGFR mutant NSCLC patients. Light grey represents tested genes that were undetected in that sample.

**Figure 5 curroncol-32-00554-f005:**
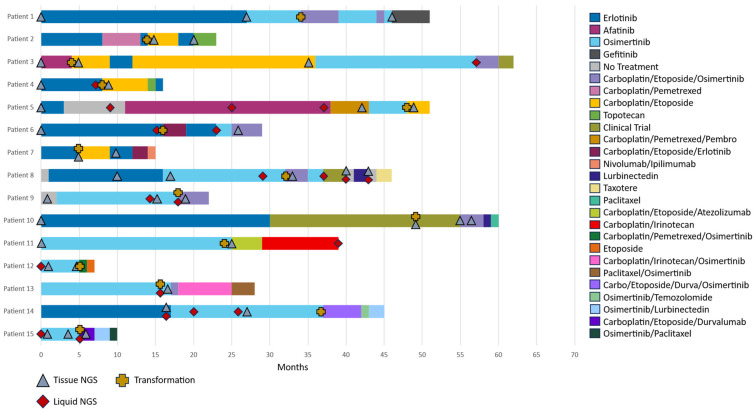
Swimmer plot demonstrating each patient’s time clinical outcomes. The yellow cross denotes the time at which transformation to SCLC occurred. The blue triangle represents timepoints at which tissue NGS was performed. The red diamond denotes timepoints at which time liquid biopsy NGS testing was performed. Each color-coded bar details the duration of treatment as characterized by the table on the right-hand side of the figure.

**Table 1 curroncol-32-00554-t001:** Demographic Information.

Patient Characteristics	N (%)
Total enrolled patients, No.	15 (100%)
Age, years, median (range)	60.0 (35.0 72.0), IQR (54.5, 66.5)
Gender	
Female	7 (46.7%)
Male	8 (53.3%)
Race	
White/Caucasian	6 (40%)
Asian	9 (60%)
Ethnicity	
Hispanic or Latino	(0%)
Not Hispanic or Latino	15 (100%)
Smoking status	
Former Smoker	5 (33.3%)
Former; Passive Exposure	1 (6.7%)
Never Smoker	8 (53.3%)
Never Smoker; Passive Exposure	1 (6.7%)
EGFR founder	
EGFR L858R mutation	3 (20%)
EGFR Exon 19 deletion	11 (73.3%)
EGFR L861Q mutation	1 (6.7%)
Median line of therapy before transformation (range)	1 (1–4)

## Data Availability

The data presented in this study are available upon request from the corresponding author due to privacy concerns.

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
