# Peer review of "Small Cell Transformation of EGFR-Mutant NSCLC Treated with Tyrosine Kinase Inhibition"

_curroncol, 2025, doi:10.3390/curroncol32100554_

Round 1

Reviewer 1 Report

Comments and Suggestions for Authors

This manuscript from Rock et al is a very well written manuscript detailing the phenotypic transformation of EGFR-mutant NSCLC treated with osimertinib. 
This manuscript adds value in clinical details for these patients. The authors highlight 15 cases, and while the number is small, and the authors acknowledge this, it is probably bigger than any other database from a single institution. 
While the genetic variations are probably worth exploring more, this was only investigated on 5 patients. 
Overall, with its limitations like sample size, and single institution, it is worthy of publication for the innovative questions and preliminary data 

Author Response

This manuscript from Rock et al is a very well written manuscript detailing the phenotypic transformation of EGFR-mutant NSCLC treated with osimertinib. 
This manuscript adds value in clinical details for these patients. The authors highlight 15 cases, and while the number is small, and the authors acknowledge this, it is probably bigger than any other database from a single institution. 
While the genetic variations are probably worth exploring more, this was only investigated on 5 patients. 
Overall, with its limitations like sample size, and single institution, it is worthy of publication for the innovative questions and preliminary data 

We would like to thank the reviewer for their meaningful feedback and agree that this manuscript is timely.

Reviewer 2 Report

Comments and Suggestions for Authors

This paper is of interest and an important contribution to the literature.  It is well thought out and makes relevant suggestions for further studies to strengthen the findings.  The tables and diagrams are clear and well constructed.  I would expand on the definition of "histologic transformation" (HC).  This paper concerns non-small-cell lung cancer (adenoca) but there is reference to small-cell lung cancer (SCLC).  Intuitively, it seems that SCLC may be an example of HC.  I have not heard of this before.  If so, are there other forms of transformations and what frequency?  A few minor points:  in the abstract, I would change "up to 50%" in line 15 to "15% - 50% (as in line 44) as the lower number is more reflective of the North American experience.  The sentence in line 237-238 seems not to be complete.  I am not sure what is meant by "GAs" in line 242.  The main weakness in this paper is clearly the small number of cases analyzed which is acknowledged throughout the manuscript.

Author Response

This paper is of interest and an important contribution to the literature.  It is well thought out and makes relevant suggestions for further studies to strengthen the findings.  The tables and diagrams are clear and well constructed.  I would expand on the definition of "histologic transformation" (HC).  This paper concerns non-small-cell lung cancer (adenoca) but there is reference to small-cell lung cancer (SCLC).  Intuitively, it seems that SCLC may be an example of HC.  I have not heard of this before.  If so, are there other forms of transformations and what frequency? 

Thank you for the feedback. We agree that this needed additional clarification and addended the manuscript to include the other possible examples of HT however the proportions are poorly defined given the rarity of each event.

A few minor points:  in the abstract, I would change "up to 50%" in line 15 to "15% - 50% (as in line 44) as the lower number is more reflective of the North American experience. 

We agree with this suggestion and have edited the manuscript accordingly.

The sentence in line 237-238 seems not to be complete. 

This has been corrected.

 I am not sure what is meant by "GAs" in line 242.

This abbreviation has been removed as it was not previously defined.  Thank you.

  The main weakness in this paper is clearly the small number of cases analyzed which is acknowledged throughout the manuscript.

We completely agree and hope to further evaluate this rare population in a multi-institutional fashion in the future.

Reviewer 3 Report

Comments and Suggestions for Authors

As attached.

Author Response

Orginality:

Histologic transformation of NSCLC to SCLC is a clinically challenging mechanism of acquired resistance, often occuring under treatment with EGFR TKIs. The present study builds upon prior HT evidence by integrating longitudinal genomic profiling from tissue and liquid biopsies.

Methodology:

The study should be regarded as a descriptive,as genomic alterations are presented as frequencies without statistical comparisons.

We completely agree with this and have noted this in the discussion.

 The retrospective design and small sample size is justified by the rarity of EGFR-mutant HT. While the precise incidence within total EGFR-mutant population cannot be calculated, this limitation is transparently discussed with suggestions for future research.

I would suggest addressing these additional methodological concerns:

-The heterogeneity of utilized NGS platforms, which introduces variability in mutation frequency estimates

Thank you for this suggestion.  We have now acknowledged this in both the methods as well as the limitations in the discussion.

-Potential overinterpretation due to wide confidence intervals.

Thank you for the suggestion.  This has also been noted in limitations.

-The method used to calculate median follow-up (reverse Kaplan–Meier?)

Thank you for this suggestion.  We have clarified in the text and methods.

-More details on instutitional protocol for germline testing

We agree that this should be expanded upon and have provided additional information regarding the protocol within the methods.

Results:

The results are presented clearly, supported by comprehensive tables and figures. However, figure 3 and 5 require refinement;

-Figure 3: While informative, the visualization is difficult to follow due to color overlap. Would benefit from clarification of Report A-B. Integration of molecular findings with treatment history or relevant clinical outcomes

Thank you for this feedback. We have provided additional clarity in the caption describing the figure.

-Figure 5: For several patients (patients 2, 7, and 13), the first presented molecular sample was visualized after HT. A clearer explanation of the figure content regarding molecular profiling time-points, or stratification of molecular data as pre- and post-transformation would enhence the validity of conclusions.

We agree with this suggestion and have edited the caption to provide additional clarity.

Implications for Research and Practice

The authors suggest that HT in EGFR-mutant NSCLC involves recurrent alterations beyond TP53/RB1, such as PIK3CA/AKT/PTEN pathway. The study provides adequate context explaining relevance of this work to current literature, ongoing efforts in combination therapies and exploration of novel therapeutic targets.

Quality of Communication

The manuscript is generally well written and logically structured. Several areas would benefit from improved clarity:

We thank the reviewer for their encouraging comments and have corrected the manuscript as requested.

-Abbreviations (e.g.,SCLC) should be defined at first mention in both the abstract and main text. We agree and this has been edited accordingly.  

-Simplify or briefly explain complex genomic terms (such as,intragenic complex re-arrengements) to increase accesibility.  

This has been defined further at the first mention.

Overall, despite the limited cohort size and methodological constraints, the study provides valuable insights into an underexplored yet clinically relevant resistance mechanism. I recommend the manuscript for publication after minor revision.

Reviewer 4 Report

Comments and Suggestions for Authors

The authors present a case series of patients with histologic transformation of SCLC and genomic characteristics of the associated tumors over time. This is a well done study and overall well presented, and would be a meaningful contribution to the field. However, there is some room for improvement in regard to presentation and discussion, please see comments below:

- It seems all cases are of transformation into small cell carcinoma. Though more rare, squamous and even sarcomatoid transformations have been reported. It may be worth briefly reviewing these in the introduction and specifying criteria in the Methods, unless the authors wish to focus solely on small cell transformation (which I would perhaps then put into the title).

Introduction

  • Line 64, I think many readers will be familiar with the concept but would still formally give brief definition and description of small cell transformation as part of the introduction
  • There are a few larger genomic studies of transformation that are probably worth reviewing and citing, for example Sivakumar et al PMID 37062002. Some of these studies bring up the idea of alternative pathways that mimic RB1 loss and other mechanistic insights that should probably be addressed

Figure 2: it is not clear what the sub columns for each patient represent: are they serial time points? If so, given important differences in sensitivity thresholds, would denote tumor vs liquid biopsy.

Figure 3: This is an interesting figure but looks a little busy.  Would consider adjusting text to minimize overlapping with the plot (e.g. move first column text to the left, removing the word “Report A” or “Report B” which can just denoted at the top or in the legend). I don’t feel strongly here but it may be better to remove the VUS which would highlight the truly significant alterations.

Figure 5: May be worth extracting out some additional observations and laying them out for the reader. For example, would be interesting to know if all patients were on a TKI at time of transformation (from my interpretation this appears to be the case but please confirm), and if not, what the last therapy was and the lag time from last exposure to TKI. Also, the number of colors is a bit overwhelming, would consider grouping into categories (such as carbo/etop+IO combinations, taxanes, maybe even all single-agent chemo). If this were done, I would still keep the various TKIs separate and separate carbo/etop from other platinum doublets

Grammar:

Line 53 “in” is italicized , similar with “altered” in line 195

Line 58, not clear what is meant by TKIs being “suppressive” in this setting

Line 174, the “to further investigate” phrase seems like you’re introducing a new analysis, might rephrase to avoid confusion

Author Response

The authors present a case series of patients with histologic transformation of SCLC and genomic characteristics of the associated tumors over time. This is a well done study and overall well presented, and would be a meaningful contribution to the field. However, there is some room for improvement in regard to presentation and discussion, please see comments below:

- It seems all cases are of transformation into small cell carcinoma. Though more rare, squamous and even sarcomatoid transformations have been reported. It may be worth briefly reviewing these in the introduction and specifying criteria in the Methods, unless the authors wish to focus solely on small cell transformation (which I would perhaps then put into the title).

Thank you for this feedback.  We have altered the title and also reported alternative examples of histologic transformation in the introduction.

Introduction

  • Line 64, I think many readers will be familiar with the concept but would still formally give brief definition and description of small cell transformation as part of the introduction
    This has been added.
  • There are a few larger genomic studies of transformation that are probably worth reviewing and citing, for example Sivakumar et al PMID 37062002. Some of these studies bring up the idea of alternative pathways that mimic RB1 loss and other mechanistic insights that should probably be addressed
    Thank you for this suggestion.  We have included information from this reference.

Figure 2: it is not clear what the sub columns for each patient represent: are they serial time points? If so, given important differences in sensitivity thresholds, would denote tumor vs liquid biopsy.

We have remade the figure and denoted tumor vs liquid biopsy as requested.

Figure 3: This is an interesting figure but looks a little busy.  Would consider adjusting text to minimize overlapping with the plot (e.g. move first column text to the left, removing the word “Report A” or “Report B” which can just denoted at the top or in the legend). I don’t feel strongly here but it may be better to remove the VUS which would highlight the truly significant alterations.

We have clarified further in the figure legend the description of the figure as per the previous reviewer. Report B on the right side illustrates all of the genomic alterations that occurred at the second genomic assessment. VUS was included in the figure as oncogenic mutations reported were sparse and the occurrence of various VUS mutations suggests other mechanisms may be involved.

Figure 5: May be worth extracting out some additional observations and laying them out for the reader. For example, would be interesting to know if all patients were on a TKI at time of transformation (from my interpretation this appears to be the case but please confirm), and if not, what the last therapy was and the lag time from last exposure to TKI. Also, the number of colors is a bit overwhelming, would consider grouping into categories (such as carbo/etop+IO combinations, taxanes, maybe even all single-agent chemo). If this were done, I would still keep the various TKIs separate and separate carbo/etop from other platinum doublets

Thank you for the suggestions.  We have included a comment regarding the time of transformation occurring solely while on TKI based therapy.

Grammar:

Line 53 “in” is italicized , similar with “altered” in line 195

This has been edited as requested.

Line 58, not clear what is meant by TKIs being “suppressive” in this setting. 

This is implying that there will remain persistent cells that can result in eventual resistance.  This has been clarified.

Line 174, the “to further investigate” phrase seems like you’re introducing a new analysis, might rephrase to avoid confusion  

Thank you for the suggestion.  We agree and have edited this accordingly.